# Financial Burden of Stroke Reflected in a Pilot Center for the Implementation of Thrombolysis

**DOI:** 10.3390/medicina56020054

**Published:** 2020-01-28

**Authors:** Diana Uivarosan, Simona Bungau, Delia Mirela Tit, Corina Moisa, Ovidiu Fratila, Marius Rus, Ovidiu Gabriel Bratu, Camelia C. Diaconu, Carmen Pantis

**Affiliations:** 1Department of Preclinical Disciplines, Faculty of Medicine and Pharmacy, University of Oradea, 410073 Oradea, Romania; diana.uivarosan@gmail.com; 2Department of Pharmacy, Faculty of Medicine and Pharmacy, University of Oradea, 410028 Oradea, Romania; mirela_tit@yahoo.com (D.M.T.); corinamoisa@hotmail.com (C.M.); 3Department of Medical Disciplines, Faculty of Medicine and Pharmacy, University of Oradea, 410073 Oradea, Romania; ovidiufr@yahoo.co.uk (O.F.); rusmariusr@yahoo.com (M.R.); 4Clinical Department 3, University of Medicine and Pharmacy “Carol Davila”, 050474 Bucharest, Romania; ovi78doc@yahoo.com; 5Department 5, University of Medicine and Pharmacy ”Carol Davila”, 050474 Bucharest, Romania; drcameliadiaconu@gmail.com; 6Internal Medicine Clinic, Clinical Emergency Hospital of Bucharest, 014461 Bucharest, Romania; 7Department of Surgical Disciplines, Faculty of Medicine and Pharmacy, University of Oradea, 410073 Oradea, Romania; pantisc@yahoo.com; 8Emergency Clinical County Hospital, 410169 Oradea, Romania

**Keywords:** health services, healthcare costs, hospitalization, stroke, thrombolysis, sustainable public health system

## Abstract

Stroke represents a serious illness and is extremely relevant from the public health point of view, implying important social and economic burdens. Introducing new procedures or therapies that reduce the costs both in the acute phase of the disease and in the long term becomes a priority for health systems worldwide. The present study quantifies and compares the direct costs for ischemic stroke in patients with thrombolysis treatment versus conservative treatment over a 24-month period from the initial diagnosis, in one of the 7 national pilot centres for the implementation of thrombolytic treatment. The significant reduction (*p* < 0.001) of the hospitalization period, especially of the days in the intensive care unit (ICU) for stroke, resulted in a significant reduction (*p* < 0.001) of the total average costs in the patients with thrombolysis, both at the first hospitalization and for the subsequent hospitalizations, during the period followed in the study. It was also found that the percentage of patients who were re-hospitalized within the first 24-months after stroke was significantly lower (*p* < 0.001) among thrombolyzed patients. The present study demonstrates that the quick intervention in cases of stroke is an efficient policy regarding costs, of Romanian Public Health System, Romania being the country with the highest rates of new strokes and deaths due to stroke in Europe.

## 1. Introduction

Stroke represents a serious illness which is extremely important from the public health point of view, implying important social and economic burdens [1]. Every year approximately 16 million cases of first stroke occur worldwide causing about 5.7 million deaths [2]. It is a major health problem both for the negative effect on the motor and cognitive abilities, and the quality of life of the survivor of the disease and his family [3,4]. After a stroke, the patient remains with a major and profound disability caused by many locomotor impairments, communication, behavioural, locomotor independence, orientation and social integration [5]. These long-term disabilities generate huge socio-economic costs [6]. The healthcare burden of stroke in European Union (EU) was approximately €20 billion in 2015, and €27 billion in 2017. In 2015, almost 72% of these expenses were represented by hospital care [7], and in 2017 a decrease of the percentage was observed; only 45% of the expanses were incurred by healthcare systems [8]. The results of the studies revealed that the costs regarding stroke (health and social care costs) / capita are associated with increases in a nation’s wealth, resulting in increased stroke-related costs [7,8]. Also, large variations in health and social care expenses for stroke have been indicated, even for those countries with similar levels of national income [8]. In 2015, in Romania, the costs / patient in case of stroke were about €8, occupying the penultimate place in the EU, ahead of Bulgaria (€7) and far from Finland, with the highest costs (€132) [7]. The overall expenditures are however, doubled, and this huge burden of stroke on the economy is supported by the society through contributions to insurance and payment of taxes but also, considerably, by the survivors of stroke and their caregivers [7,8].

The 2015 statistical reports showed that Romania is the country with the highest rates of new strokes and deaths due to stroke in Europe, stroke being the second leading cause of mortality—21.64% and disability—11.34%. There are 61,500 strokes are registered annually, most of them (55,000) of which are ischemic accidents [7,9].

Although there are favourable studies to implement thrombolytic therapy, the implementation of specific treatment is still a challenge worldwide. In 2004, 8 years after the approval of the recombinant tissue plasminogen activator (rtPA), both doctors and specialized centres had difficulties in adapting to the use of thrombolytic treatment, the administration of rtPA being possible in only 1–2% of the patients eligible for treatment [10,11]. The main disadvantage of this therapy is the narrow therapeutic window (4.5 h), the time period in which the patient must be diagnosed, transported to a specialized unit, investigated and finally treated, through the applied therapy. The shorter the time from stroke onset, the shorter the therapeutic efficacy. Intravenous thrombolysis is initiated only if the time interval from the first symptoms of stroke to the time of administration is less than 4.5 h [12].

In Romania, the situation is still at the beginning. The doctors and hospitals are not fully prepared to provide adequate medical care in a very short time to a patient who has had an acute stroke. Between 2008 and 2014, the average annual thrombolysis rate per 100,000 population was 0.1 [13]. In 2018, only 14 hospitals performed this procedure, so over 74% of the counties in the country did not have, at the time, any hospital adequately equipped to be able to apply the thrombolysis procedure, while the other counties had only one hospital prepared for the procedure. In 2018, the Romanian Neurology Society board founded the National Program of Priority Actions in the Interventional Treatment of Patients with Acute Cerebral Vascular Accident (PA-CVA) following the results of a pilot study on the mechanism and way of action of thrombolysis treatment in patients with acute ischemic stroke (AIS). The hospitals having the required facilities and doctors prepared and willing to participate in implementing the thrombolysis treatment were included in the program [10]. It was only at the end of 2019 that at least one medical center was established in each county of Romania where thrombolysis is performed.

The present study quantifies and compares the direct costs for ischemic stroke in patients with thrombolysis treatment versus conservative treatment, over a 24-month period from the initial diagnosis, in one of the 7 national pilot centers for the implementation of thrombolytic treatment, located in Oradea, Romania. The results of this study can be generalized for all the pilot centers existing in the country, going through the following reasons: all the other centers included in the pilot program are similar, being integrated in public hospitals; all expenditures are covered by the national health system; the costs regarding the hotel regime, the medication and the consumables are almost identical; the legal frame is the same.

## 2. Materials and Methods

### 2.1. Data Collection

Retroactive analysis of the medical diagram using information obtained from the Oradea County Clinical Emergency Hospital, Neurology section, one of the 7 pilot centers nationwide, which applies the procedure of intravenous thrombolysis in AIS at the time of the study. The study included patients over 18 years old, hospitalized with ischemic stroke diagnosis, diagnostic code I63.0-9. Patients receiving thrombolytic treatment were selected according to procedure code H13901. Each patient was monitored for 24-months from the first diagnosis. The direct costs at the initial hospitalization with ischemic stroke diagnosis and at the re-hospitalizations in the next 24-months were analyzed for the patients to whom the thrombolysis procedure was applied and compared with those for the patients with conservative treatment. In total, 831 patients with AIS and aged up to 75 years hospitalized between September 2016 and September 2017 were included in the research. Of these, both deaths (those patients who died during the first hospitalization and could not be monitored until the end of the study)—96 patients, as well as patients transferred to other hospitals— 89 patients were eliminated from the study, finally remaining 646 patients. The demographic characteristics of all patients were registered. To eliminate the influence of age difference or other demographic criteria, from the group with patients with conservative treatment it was selected a sample with similar characteristics to those from the group with thrombolyzed patients. Thus, the study included 3 patient groups: Group I—55 patients with thrombolysis, Group II—591 patients without thrombolysis, and Group III—248 patients without thrombolysis (the sample selected from group II).

The study was approved by the Ethical Commission of the Faculty of Medicine and Pharmacy (Ethical no. 10841/28.10.2019) of the University of Oradea, Oradea, Romania.

### 2.2. Cost Analysis

The total costs of hospitalization during the stay, for the first hospitalization with AIS diagnosis, as well as the costs of hospitalization during the next 24-months were analysed from the economic point of view. The costs for every type of expenses were registered: costs for medication, medical supplies, medical analyses, imaging and other investigations, costs for stationary hospital stay and intensive care unit (ICU) stay. All costs were calculated in national currency and converted into euro, at the average exchange rate valid in the years of the study period. Hotel costs include accommodation and meals. During the period studied, the average cost of a day of hospitalization (hotel cost) per Neurology section was €68.47, and for ICU was €205.190.

### 2.3. Statistical Analysis

The statistical analysis was performed with SPSS 20. In order to apply the t test for this research, the data distribution was tested by applying the Kolmogorov–Smirnov test; a normal distribution resulted (*p* > 0.05). All the average parameter values, standard deviations, frequency ranges, and statistical significance tests were calculated by using the Student method (t test and χ^2^), considering the *p* < 0.05 value as statistically significant.

## 3. Results

Only 6.6% of the patients admitted to the hospital during the study period were eligible for thrombolytic treatment. The demographic characteristics of the all patient are presented in Table 1. Between group I and II, there are statistically significant differences, both in terms of age and the environment of origin.

The duration of the first hospitalization was 5.56 days for the group I patients, about 1.3 times higher (7.42 days) for the group II patients, and about 1.2 times higher (6.48 days) for the group III. In the ICU, 16.36% (*N* = 9) of the thrombolyzed patients and 33.5% (*N*= 198) of the non-thrombolyzed patients were admitted. The ICU stay ranged from 1 to 10 days, with a mean of 0.51 days in the group I, 1.66 and 0.97 days in the groups I and II (Table 2).

The main cost categories are presented in Table 3. It can be observed that there are no significant differences between the total average costs / day of hospitalization for rt-PA treatment and the costs for conservative treatment (131.33 vs. 128.74 and 143.11). However, due to the reduction of the length of hospitalization time, especially of the number of days spent at the ICU, the average total costs are significantly lower (*p* < 0.001) in the case of thrombolyzed patients (730.20 vs. 955.27 and 927.39).

Regarding the re-hospitalizations in the next 2 years after stroke, the percentage of patients with re-hospitalizations was significantly lower in the case of the thrombolyzed patients than in the non-thrombolyzed patients (1.82% vs. 11.00%, and 14.11%). It is observed that 1.82% of the patients with thrombolysis had 2 re-hospitalizations (with an average hospitalization length of 5.82 days), while in the groups with conservative treatment 8.29% and 10.08% had 2 re-hospitalizations and 2.71% and 4.03% had 3 hospitalizations with an average hospitalization length of 8.36 and 6.74 days. The mean duration of hospitalization on ICU was longer in patients with conservative treatment, in the acute phase (2.72 and 1.06 days vs. 0.58 days) (*p* < 0.001) (Table 4).

The risk analysis showed that the risk of rehospitalization is significantly higher for patients with conservative treatment in both groups (OR = 3.275, 95% CI: 0.777–13.755, *z* = 1.60, *p* = 0.105, respectively OR = 4.3545, 95% CI: 1.015–18.0683, *z* = 1.980, *p* =0.048).

Considering re-hospitalization, the average daily costs were significantly lower in patients with thrombolysis (127.64) than in patients who underwent conservative treatment at first hospitalization (143,79 and 173.63). The total average cost of the thrombolyzed patients was €742.87, and of the non-thrombolized patients €1202.11 and 1170.32 (* *p* < 0.001), the main cost categories in the case of re-hospitalization being presented in Table 5. The complete characterization of the variables in the study (descriptive statistics and *p*-values) is shown in Table 6.

## 4. Discussion

This study focuses on the cost-effectiveness of thrombolysis in AIS, compared to that of conservative treatment. Two main categories of costs were assessed: at the first hospitalization and in the first 24-months thereafter, for problems related to vascular accident. The results indicated that over 70% of the expenses are represented by the hotel costs. The significant reduction of the hospitalization period, especially of the days in ICU, resulted in a significant reduction of the total average costs in the patients with thrombolysis both at the first hospitalization and for the subsequent hospitalizations, during the period followed in the study. It was also found that the percentage of patients who were re-hospitalized within the first 24-months after stroke was significantly lower among thrombolyzed patients. This fact suggests a much better recovery of these patients.

Previous studies have shown that the administration of tissue plasminogen activator increases the recovery of stroke symptoms by up to 50% [14], with a decrease of serious complications [15,16]. However, studies show that only 3, up to 8.5% of eligible patients receive rtPA [17]. Ideally, more than 40% of patients eligible for thrombolytic treatment should receive this treatment, however that is impossible to achieve due to several factors: insufficient funding for these facilities and for the necessary personnel, poor awareness of stroke symptoms, both by patients and by the patients’ families, the fear of physicians to be legally responsible for treatment administration and validation of patient eligibility [18]. From March 2012 (when this type of treatment started in Romania) until 2018, at the Oradea County Emergency Hospital, thrombolysis was used in 200 cases of ischemic stroke, representing 3% of total ischemic stroke cases [10]. Results of this study have shown a percent of 6.6% patients with AIS registered in 2017, indicating an increase in the thrombolysis rate at the hospital level. However, there are a number of reasons that make administering this treatment difficult. Of these we mention: the limited time window available for the application of the treatment, the problematic coordination of the specialists taking into account the limited time they have at their disposal, the doctors’ concern for the bleeding complications that may occur, the laborious process of obtaining the patient’s consent [10]. Several studies have reported different percentages of using thrombolytic treatment. A higher percentage was identified in other similar studies, 6% in India and 7% in the USA [19,20].

In 2019, the national program expanded, so that at the beginning of the year, over 95% of the counties in Romania have at least one hospital prepared to apply the procedure. At the same time, a series of measures, procedures and indications regarding the care of the patient with acute stroke were applied in order to increase the efficiency of the emergency system, the rapid recognition of the stroke signs, the formation of rapid intervention teams in order to increase the number of patients to benefit from thrombolysis and enlargement of the therapeutic window [10]. One of the reasons why patients that have suffered an acute stroke are not given rtPA treatment is that they are not informed about the risk factors and symptoms of stroke, so they are not aware of the disease and do not arrive in time in the emergency department to be administered with the treatment. Another obstacle that limits the use of plasminogen activator is the fear of physicians of the frequency of serious side effects [18,21].

Stroke is a condition that can have multiple effects on the general state of the patient, therefore the care can raise ethical problems. Furthermore, stroke influences the life of the caregivers. The process of treating patients with stroke is carried out in several stages being a difficult task for health professionals. The opinion of all people involved regarding stroke is that the care involves various ethically sensitive situations. Saving the life of a patient must be a priority over any economic calculation. Health professionals must fight for the life of each patient. If the constant refusal of a patient to follow the treatment affects the society and the individual, there must be taken appropriate and proficient measures to ensure the intervention. In spite of the various perspectives concerning the medical care pattern, investigators reveal that the physician should not make use of the liberty principle in formulating competence or taking decisions that have negative effects on patients or society. Before taking any clinical measures, healthcare professionals have to make sure they are acting according to medical ethics. Options to drop vital treatment have to take into account the patient’s desire along with the analyses comparing the burden and benefit [22].

Despite the evidence that long-term costs are substantial, few studies have quantified short-term and long-term direct costs [23,24]. Studies that systematically evaluated readmission after stroke have shown that patients with a more severe form of the disease have higher readmission rates both at 30 and 90 days [25,26]. Also, re-hospitalizations are more frequent in the elderly, one of the studies indicating relatively high readmission rates, 25–39% within 30 days [25]. In another recent study on 307,887 patients with insurance in the public health system who suffered an ischemic stroke, 14.4% of patients had readmissions within 30 days after discharge [25,26]. The readmission rate adjusted at 30 days, reported by Fonarow et al. [27] in their study, was 14.1%, which was more than double at 90 days, 29.2%. One other research shows that 28% of the analysed patients had readmissions, within 30 days, the majority of readmissions being in patients who had sequelae related to stroke [25].

The present study reports readmissions over 24-months, only for stroke-related causes, and indicates a rate of about 14% for patients without thrombolysis and less than 4% for the other group of patients. In addition to this significant difference regarding the number of hospitalizations, there is also a significant decrease in the length of hospitalization, especially of the number of days spent at the ICU for the thrombolyzed patients. All these differences substantially reduce long-term costs in thrombolyzed patients.

Interestingly, in the published studies that estimate total costs within one year after ischemic stroke in the US, the costs attributed to initial hospitalization are decreasing. Taylor et al. [28] found that 70% of the costs in the first year were attributed to the initial hospitalization while other studies indicate that 62%, respectively 50% of the costs were attributed to this hospitalization [25,29]. The results obtained indicate that a percentage below 50% is attributed to the expenses for the first hospitalization, both for patients with thrombolysis and for those with conservative treatment, and a significantly reducing of total direct costs on the thrombolyzed patients.

Our work reveals that quick intervention in cases of stroke is an efficient policy regarding costs of the Romanian Public Health System. Therefore, more studies are needed to evaluate the total direct and indirect costs for a more accurate assessment and clarification of the implications of thrombolytic treatment.

## 5. Conclusions

Using the rtPA treatment in the 4.5 hours’ time window following stroke influences the evolution of the disease. It is to be mentioned that rtPA treatment represents the singular efficient intervention currently available replacing the classic treatment. The procedure cuts down the overall costs, decreasing the hospitalization and rehabilitation period and particularly it may decrease the indirect costs (absenteeism, productivity loss, premature death) with important socioeconomic impact.

## Figures and Tables

**Table 1 medicina-56-00054-t001:** Demographic characteristics.

Characteristics	Group I		Group II		*p*	Group III		*p*
	No.	%	No.	%		No.	%	
**Gender**								
Women	26	47.27	288	48.73	0.836	123	49.60	0.675
Men	29	52.73	303	51.27		125	50.40	
Total	55	100.00	591	100.00		248	100.00	
**Age Group (Years)**								
<40	5	9.09	8	1.35	0.031	22	8.87	0.629
41–50	16	29.09	56	9.48		76	30.65	
51–60	18	32.73	84	14.21		84	33.87	
61–70	12	21.82	242	40.95		49	19.76	
71–75	4	7.27	201	34.01		17	6.86	
Average Age	53.56 ± 7.46		64.83 ± 9.21			54.15 ± 8.32		
**Environment**								
Urban	38	69.09	270	45.69	<0.001	57.26	57.26	0.107
Rural	17	30.91	321	54.31		42.74	42.74	

**Table 2 medicina-56-00054-t002:** Average length of hospitalization—first hospitalization.

Group	Neurology	ICU	Total
I	5.05 ± 1.21	0.51 ± 0.12	5.56 ± 1.24
II	5.76 ± 2.41	1.66 ± 0.88	7.42 ± 2.34
*p*	0.031	<0.001	<0.001
III	5.51 ± 1.62	0.97 ± 0.48	6.48 ± 1.87
*p*	0.048	<0.001	<0.001

ICU: intensive care unit; III: the sample of patients selected from group II;

**Table 3 medicina-56-00054-t003:** The main cost categories—first hospitalization.

Cost Categories	Minimum	Maximum	Mean	Standard Deviation	%
**Group I (*N* = 55)**					
Medicines	78.85	230.15	65.38	26.28	8.95
Sanitary materials	2.21	28.05	9,50	5,08	1.30
Medical analysis	24.30	80.62	39.54	9.94	5.41
Imaging and other investigations	29.82	223.52	97.18	47.71	13.31
Hotel costs	194.15	1335.84	518.61	188.62	71.03
**Total**	314.74	1632.69	730.20 *	236.78	100.00
**Group II (*N* = 591)**					
Medicines	25.84	664.16	90.56	52,79	9.48
Sanitary materials	2.21	65.60	14.80	9.50	1.55
Medical analysis	19.22	214.91	40,20	13.03	4.21
Imaging and other investigations	23.19	337.93	104.25	45.72	10.92
Hotel costs	222.86	1811.37	705.47	279.40	73.84
**Total**	312.09	2387.63	955.27 *	356.49	100.00
**Group III (*N* = 248)**					
Medicines	23.22	646.16	88.55	50.12	9.55
Sanitary Materials	1.88	60.46	13.22	8.63	1.43
Medical Analysis	17.64	202.17	38.86	16.41	4.19
Imaging and Other Investigations	18.81	341.00	98.88	46.66	10.66
Hotel Costs	193.12	1868.46	687.88	251.37	74.17
**Total**	300.1	2471.33	927.39 *	361.21	100.00

* *p* < 0.001.

**Table 4 medicina-56-00054-t004:** Number of hospitalizations and average length of stay.

No. of Hospitalizations	Group I		Group II		Group III	
	Patients					
	No.	%	No.	%	No.	%
One	54	98.18	526	89.00	213	85.89
Two	1	1.82	49	8.29	25	10.08
Three	0	0.00	16	2.71	10	4.03
**Average length of hospitalization (days)**						
Neurology	5.24 ± 1.55		5.64 ± 3.22		5.68 ± 2.14	
ICU	0.58 ± 0.30 *		2.72 ± 1.41		1.06 ± 0.63	
Total	5.82 ± 1.28 *		8.36 ± 3.46		6.74 ± 1.84	

* *p* < 0.001.

**Table 5 medicina-56-00054-t005:** The main cost categories, in €, with re-hospitalizations.

Cost categories	Minimum	Maximum	Mean	Standard Deviation	%
**Group I (*N* = 55)**					
Medicines	76.72	233.59	87.46	41.04	11.77
Sanitary Materials	2.15	27.29	9.45	5.16	1.27
Medical Analysis	23.64	85.31	38.89	9.45	5.24
Imaging and Other Investigations	29.01	224.78	96.92	45.77	13.05
Hotel Costs	192.11	1299.67	510.15	187.60	68.67
**Total**	315.03	1588.48	742.87*	230.79	100.00
**Group II (*N* = 591)**					
Medicines	25.14	1032.13	159.02	85.10	13.23
Sanitary Materials	2.15	194.69	46.20	19.12	3.84
Medical Analysis	18.70	365.75	56.09	18.70	4.67
Imaging and Other Investigations	22.56	642.74	135.17	67.91	11.24
Hotel Costs	214.89	3181.05	805.63	286.24	67.02
**Total**	303.64	4327.50	1202.11	657.78	100.00
**Group III (*N* = 248)**					
Medicines	26.02	1132.22	142.12	80.87	12.14
Sanitary Materials	3.12	189.18	40.22	18.88	3.44
Medical Analysis	19.55	351.82	50.26	17.95	4.29
Imaging and Other Investigations	20.27	614.11	137.5	65.40	11.75
Hotel Costs	216	3062.2	800.22	262.55	68.38
**Total**	310.11	4022.70	1170.32	489.11	100.00

* *p* < 0.001.

**Table 6 medicina-56-00054-t006:** The complete characterization of the variables (group I vs. group II and III).

Evaluation	Group	Mean Difference	Standard Error	Sign	95% CI	
					Lower	Upper
**Age**	II	11.270	1.280	0.000	8.758	13.783
	III	0.590	1.218	0.629	−1.807	2.987
**First Hospitalization—Length of Hospitalization**						
Neurology	II	0.710	0.329	0.031	0.064	1.356
	III	0.460	0.232	0.048	0.004	0.816
ICU	II	1.150	0.119	0.000	0.917	1.383
	III	0.460	0.065	0.000	0.332	0.588
Total	II	1.860	0.320	0.000	1.232	2.488
	III	0.920	0.264	0.000	0.3999	1.440
Total cost	II	225.070	33.343	0.000	159.597	290.543
	III	225.070	60.399	0.000	125.891	324.249
**Total Hospitalization—Length of Hospitalization**						
Neurology	II	0.400	0.439	0.363	−0.462	1.262
	III	0.440	0.305	0.150	−0.160	1.040
ICU	II	1.150	0.119	0.000	0.917	1.383
	III	0.480	0.087	0.000	0.309	0.652
Total	II	1.860	0.320	0.000	1.232	2.488
	III	0.920	0.261	0.001	0.406	1.434
Total cost	II	459.240	89.256	0000	283.972	634.508
	III	319.350	82.587	0000	156.828	484.872

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
