# Peer review of "Financial Burden of Stroke Reflected in a Pilot Center for the Implementation of Thrombolysis"

_medicina, 2020, doi:10.3390/medicina56020054_

Round 1
Reviewer 1 Report
Introduction is too long, please short it especially in the first part. From line 41 to 76 there should be only one paragraph left. This paragraph should simply state that stroke is a subjective (patient-level) and objective (society-level) burden. Also state comparable cost in Europe countries for dealing with patient in-hospital and afterwards. There are simply too many words and almost no data.
Line 83- please state incidence and prevalence of stroke (ischemic/hemorrhagic) in Romania. This is a critical reference. You mentioned only PDF document from one Web page.
It is questionable whether this report could be generalized since there is data only from singular center (of 7 participating).
Why were deaths eliminated?
Please state whether data analysed followed normal distribution or not, also state results of normal-distribution tests.
It would be of help when authors stated firstly average year of stroke patients in Romania so we can compare with this data results of present study.
Thromoblysed patients were unusually young as stated by authors. “Conservative” patients were significantly older – this represents major bias of the study. A lot of information for selecting the patients is not revealed in the paper, so final judgment can not be made. Simply to put: “The older the patient, the more frequent the associated pathology, the patient being ineligible for thrombolysis” is not acceptable. Do we know which these factors were? There is so few concomitant diseases that are exclusion criteria for thrombolysis!
Please explain the shortcut ATI in Table 2.
The formatting of all tables is unusual and difficult to interpret. Please follow standard reporting guidelines for table formatting!
Did patient stayed in Stroke Unit of Intensive care unit (ICU)? It is highly unusual that stroke patients are treated in such great numbers in ICU. Please explain. How many patients stayed in ICU? All of them? Please state the numbers.
There are no caregiver or rehabilitation costs reported. Please explain why.
There are no clinical outcome data reported. Please explain why.
In Discussion, line 216-217 is not relevant any more, study cited was from 2005. Please delete.
Sincerely, the study is of interest only to small group of Romanian neurologist and could not be (see age bias) generalized.
The facts mentioned regarding costs could be generalized only when authors report differential costs from Romanian health care system (hotel stay) with Europe average. They mentioned only briefly. When we say that medical/equipment cost is almost the same Europe-wide, then only factor of interest are local (Romanian-specific) health care costs. It is no question whether or not to apply thrombolysis. The only interesting question is whether cost of medication/equipment is of benefit to local system. From reported data, first hospitalization reduced costs for 13%, compared to conservative treatment. However, the authors should know that patients treated with thrombolysis were younger.
It is simply known that thrombolysis is cost-effective treatment and should not be proved anymore in the studies. The study is severely biased with age difference. It is reasonable to conclude that younger patients are less prone to stay in hospital longer.
Discussion is too long and should be focused on study results.
Author Response
Journal: Medicina
Manuscript: medicina-671441
We are very thankful to the Editor/Reviewers for their notes; we have carefully read the comments and have revised / completed the manuscript accordingly. Our responses are given in a point-by-point manner below, as well all the changes to the manuscript are highlighted in red.
Reviewer 1
Introduction is too long, please short it especially in the first part. From line 41 to 76 there should be only one paragraph left. This paragraph should simply state that stroke is a subjective (patient-level) and objective (society-level) burden. Also state comparable cost in Europe countries for dealing with patient in-hospital and afterwards. There are simply too many words and almost no data.
Thank you for your recommendations. We have restructured the Introduction section according to your suggestions. We hope that the new form of this section is acceptable for you.
Line 83- please state incidence and prevalence of stroke (ischemic/hemorrhagic) in Romania. This is a critical reference. You mentioned only PDF document from one Web page.
The data on the incidence and prevalence of stroke in Romania presented in the Introduction section are provided by reports of the Stroke Alliance for Europe [7,9]
It is questionable whether this report could be generalized since there is data only from singular center (of 7 participating).
We considered that the results of this study can be generalized for all the pilot centers existing in the country, going through the following reasons: all the other centers included in the pilot program are similar, being integrated in public hospitals; all expenditures are covered by the national health system; the costs regarding the hotel regime, the medication and the consumables are almost identical; the legal frame is the same.
Why were deaths eliminated?
Only those patients who died during the first hospitalization and could not be monitored until the end of the study were eliminated.
Please state whether data analysed followed normal distribution or not, also state results of normal-distribution tests.
In order to apply the t test for this research, the data distribution was tested by applying the Kolmogorov – Smirnov test; a normal distribution resulted (p>0.05). All the average parameter values, standard deviations, frequency ranges, and statistical significance tests were calculated by using the Student method (t test and c2), considering the p<0.05 value as statistically significant.
It would be of help when authors stated firstly average year of stroke patients in Romania so we can compare with this data results of present study. Thromoblysed patients were unusually young as stated by authors. “Conservative” patients were significantly older – this represents major bias of the study. A lot of information for selecting the patients is not revealed in the paper, so final judgment can not be made. Simply to put: “The older the patient, the more frequent the associated pathology, the patient being ineligible for thrombolysis” is not acceptable. Do we know which these factors were? There is so few concomitant diseases that are exclusion criteria for thrombolysis!
We are sorry for the wrong formulation we have given up. Because we wanted to validate the result obtained on the initial groups, from the group of patients with conservative treatment we selected a sample with similar characteristics. Following the application of statistical tests, on the 2 groups (the group with thrombolysed patients and the sample obtained from the other group), we obtained significant differences (as in the first submitted version of this study), which strengthens the initial conclusion. The issues that determine a low rate of thrombolysed patients, in Romania, in general, and at the level of the studied hospital are detailed in the Discussions section.
Please explain the shortcut ATI in Table 2.
We apologize for the inattention, it's about the ICU. We have corrected.
The formatting of all tables is unusual and difficult to interpret. Please follow standard reporting guidelines for table formatting!
We are sorry for the confusion; we respected the journal's requirements. We reshaped them as much as we could, taking into account the changes and the new data added to them.
Did patient stayed in Stroke Unit of Intensive care unit (ICU)? It is highly unusual that stroke patients are treated in such great numbers in ICU. Please explain. How many patients stayed in ICU? All of them? Please state the numbers.
Sorry we missed this data. Yes, this is the Stroke Unit of Intensive care. We added in the Results section the number of patients who stayed in the ICU. To clear this aspect, the following text was added in the manuscript.
In the ICU, 16.36% (n=9) of the thrombolysed patients and 33.5% (n=198) of the non-thrombolysed patients were admitted.
There are no caregiver or rehabilitation costs reported. Please explain why.
Our study is a retrospective one and was performed in an emergency hospital, where there is no rehabilitation department. In addition, the purpose was to study / evaluate the direct costs for patients with AIS.
There are no clinical outcome data reported. Please explain why.
The impossibility to differentiate between every single patient and its comorbidities is a limitation of this study; this limitation caused the lack of analysis of the clinical data of the patients. However, we can say with certainty that, from the data recorded in the patient observation sheets, all the outpatients included in the study had an improved state of health.
In Discussion, line 216-217 is not relevant any more, study cited was from 2005. Please delete.
Thanks for recommendation, we deleted it.
Sincerely, the study is of interest only to small group of Romanian neurologist and could not be (see age bias) generalized.
After consulting the studies / publications in the field, in Romania, and from our knowledge, this study is the only one in Romania that evaluates the direct costs of stroke, comparatively. At first glance it may seem of interest only to the Romanian neurologist but given the lack of reporting of this type of data in Romania, as well as in the countries of Eastern Europe, in general, we consider that this data may be of greater interest than a national one. In a comprehensive study on stroke costs in Europe for 2017 (published in 2019), all types of costs were estimated for Romania only by reference to other countries in the area, precisely because of the lack of concrete data [8].
The facts mentioned regarding costs could be generalized only when authors report differential costs from Romanian health care system (hotel stay) with Europe average. They mentioned only briefly. When we say that medical/equipment cost is almost the same Europe-wide, then only factor of interest are local (Romanian-specific) health care costs.
The recent study mentioned above [8] indicates large variations in health and social care costs for stroke, even for countries with the same level of national income. The differences arise in both emergency care, hospital care and medications. As mentioned above, we consider that these data from Romania are of wider interest.
It is no question whether or not to apply thrombolysis. The only interesting question is whether cost of medication/equipment is of benefit to local system. From reported data, first hospitalization reduced costs for 13%, compared to conservative treatment. However, the authors should know that patients treated with thrombolysis were younger.
Thanks for the comments. You are right, we focused only on the main objective of this study - to evaluate the direct costs in stroke according to the treatment followed and we ignored the importance of applying this treatment. We completed the manuscript with a few paragraphs in which we outlined the importance of thrombolysis and the evolution of this treatment in the studied hospital.
It is simply known that thrombolysis is cost-effective treatment and should not be proved anymore in the studies. The study is severely biased with age difference. It is reasonable to conclude that younger patients are less prone to stay in hospital longer.
As we have already mentioned, the differences in stroke costs between EU countries are significant; we consider that each case is particular and that this study in Romania is of particular interest through the data presented. New data added to the study have shown that age differences do not significantly influence study outcomes.
Discussion is too long and should be focused on study results.
Thanks for the observation, the Discussion section has been shortened
Reviewer 2 Report
I thank the authors for their effort they put into this nice piece of research. The authors are addressing an important issue in the field of stroke care.
Some comments:
I actually prefer the term 'thrombolysis' instead of 'fibrinolysis', as the first term is more frequently used in a clinical setting. Maybe, the authors can consider adjusting the wording accordingly. I suggest the authors to have a careful look at the manuscript once again, since English language and flow of language can be improved. If changed, the quality of this manuscript can be further improved. The worldwide death rates should be exactly the same when stated twice. The time window for IVT from symptom onset to treatment should be listed as 4.5 hours. I suggest to omit the '3 hours' and to be clear about what 4.5 mean. Introduction, line 79/80, please, be clear, about potentially eligible IVT patients. The actual numbers are higher than 1-2%. But the issues, leading to such low numbers are crucial. Conclusion: I would speak about 'cutting down the overal costs - as IVT and acute care are quite costly'. Would it be possible to include the ethical issues about 'safeing people from dying but surviving disabled' or 'letting people die, since it is cheaper'? Would it be possible, to show some Odds ratios/Confidence interval analyses as well? As far as I can see, the data would be available and I think, These parameters would further improve the quality of the manuscript.Author Response
Journal: Medicina
Manuscript: medicina-671441
We are very thankful to the Editor/Reviewers for their notes; we have carefully read the comments and have revised / completed the manuscript accordingly. Our responses are given in a point-by-point manner below, as well all the changes to the manuscript are highlighted in red.
Reviewer 2
I actually prefer the term 'thrombolysis' instead of 'fibrinolysis', as the first term is more frequently used in a clinical setting. Maybe, the authors can consider adjusting the wording accordingly.
Thanks for your recommendation. We have replaced in the text the term fibrinolysis and all derived terms, with the term thrombolysis.
I suggest the authors to have a careful look at the manuscript once again, since English language and flow of language can be improved. If changed, the quality of this manuscript can be further improved.
Thank you for recommendation. English has been revised and improved.
The worldwide death rates should be exactly the same when stated twice.
Thanks for the observation. We kept only one reference to the mortality rate; at Reviewer 1's request, part of the text in the Introduction section was deleted.
The time window for IVT from symptom onset to treatment should be listed as 4.5 hours. I suggest to omit the '3 hours' and to be clear about what 4.5 mean.
As you have recommended, we kept the time window for IVT from symptom onset to treatment for 4.5 hours and clarified what this represents.
Introduction, line 79/80, please, be clear, about potentially eligible IVT patients. The actual numbers are higher than 1-2%. But the issues, leading to such low numbers are crucial.
The 1-2% percentage was valid for 2004, 8 years after the approval of the recombinant tissue plasminogen activator (rtPA). The issues that determine a low rate of thrombolysed patients, in Romania, in general, and at this hospital level are detailed in the Discussion section.
Conclusion: I would speak about 'cutting down the overal costs - as IVT and acute care are quite costly'. Would it be possible to include the ethical issues about 'safeing people from dying but surviving disabled' or 'letting people die, since it is cheaper'? Would it be possible, to show some Odds ratios/Confidence interval analyses as well? As far as I can see, the data would be available and I think, These parameters would further improve the quality of the manuscript.
Thanks for your recommendations, as you suggested we presented Odds ratios / Confidence interval analyses in the Results section.
The risk analysis showed that the risk of rehospitalization is significantly higher for patients with conservative treatment in both groups (OR=3.275, 95% CI: 0.777-13.755, z=1.60, p=0.105, respectively OR=4.3545, 95% CI: 1.015-18.0683, z=1.980, p=0.048).
We also added information about ethical issues. Please see below.
Stroke is a condition that can have multiple effects on the general state of the patient, therefore the care can raise ethical problems. Furthermore, stroke influences the life of the caregivers. The process of treating patients with stroke is carried out in several stages being a difficult task for health professionals. The opinion of all people involved regarding stroke is that the care involves various ethically sensitive situations. Saving the life of a patient must be a priority over any economic calculation. Health professionals must fight for the life of each patient. If the constant refusal of a patient to follow the treatment affects the society and the individual, there must be taken appropriate and proficient measures to ensure the intervention. In spite of the various perspectives concerning the medical care pattern, investigators reveal that the physician should not make use of the liberty principle in formulating competence or taking decisions that have negative effects on patients or society. Before taking any clinical measures, healthcare professionals have to make sure they are acting according to medical ethics. Options to drop vital treatment have to take into account the patient's desire along with the analyses comparing the burden and benefit.